# A white-to-opaque-like phenotypic switch in the yeast *Torulaspora microellipsoides*

Cedric A. Brimacombe [1,2], Thomas Sierocinski[1] & Matthew S. Dahabieh[1✉]

*Torulaspora microellipsoides* is an under-characterized budding yeast of the Saccharomycetaceae family that is primarily associated with viticulture. Here we report for the first time to our knowledge that *T. microellipsoides* undergoes a low-frequency morphological switch from small budding haploid (white) yeast to larger, higher ploidy (opaque) yeast. Comparison of transcriptomes by mRNA-seq revealed 511 differentially regulated genes, with white cells having greater expression of genes involved in stress resistance and complex carbohydrate utilization, and opaque cells up-regulating genes involved in ribosome biogenesis. Growth assays showed that white cells are physiologically more resistant to stationary-phase conditions and oxidative stress, whereas opaque cells exhibited greater cold tolerance. We propose that phenotypic switching in *T. microellipsoides* is an ecological adaptation, as has been suggested for similar morphological switching in distantly related species like *Candida albicans*, and we propose that this switching is a more broadly utilized biological strategy among yeasts than previously thought.

[1] Renaissance BioScience Corporation, 410-2389 Health Sciences Mall, Vancouver, BC V6T 1Z3, Canada. [2] Department of Microbiology and Immunology, The University of British Columbia, 2350 Health Sciences Mall, Vancouver, BC V6T 1Z3, Canada. ✉email: mdahabieh@renaissancebioscience.com

Many microorganisms undergo phenotypic switches, wherein a single genotype can generate multiple different phenotypes, often in response to environmental changes. This phenomenon can enable rapid adaptation of these organisms to fluctuating environments and, ultimately, can provide competitive advantages in specific ecological niches. An example of this includes conformational changes of various prion proteins in the yeast *Saccharomyces cerevisiae* that result in changes to both individual and population-level behaviours, such as carbon metabolism and flocculation, respectively[1–3]. Additionally, different transcriptional programs are associated with morphology changes in many unicellular fungi. A species of particular interest is *Candida albicans*, which undergoes several phenotypic switches including the yeast to hyphal and white-to-opaque[4,5] switches. These phenotypic switches are stabilized transcriptionally[6], are stable for many generations, and have been extensively studied with regard to mechanisms of switching and regulation of the lifestyle of *C. albicans*[5,7].

The white-to-opaque switch in *C. albicans* is one of the most studied epigenetic switches in fungi, with the transition between these two distinct budding yeast phenotypes thought to be exclusive to *C. albicans* and the related CTG clade species *Candida dubliniensis* and *Candida tropicalis*. *C. albicans* exists primarily as a budding yeast that forms shiny white colonies on agar plates; however, it can spontaneously and reversibly switch to the opaque colony morphology, wherein colonies are flatter, have a darker, dull colour, and individual cells are larger and elongated[5]. This switch is thought to have evolved as an adaptation to life in mammalian hosts and, accordingly, white and opaque cells differ dramatically in their gene expression profiles, mating competency, virulence, and metabolic properties, with each morphology thought to have specific advantages in different host niches due to the differing properties of each cell type[6,8].

In ascomycetes, phenotypic switching has historically been studied in only a handful of species and isolates, most of which are domesticated lab strains of human pathogens. A number of these yeasts undergo a yeast-hyphae transition; however, white-to-opaque-like switching has not been observed outside of *Candida* species and has primarily been studied in relation to commensalism and pathogenesis of humans[9]. To our knowledge, little if any, research has been reported on the characterization of phenotypic switching in non-canonical, non-pathogenic yeast species, including the *Zygosaccharomyces/Torulaspora* (ZT) clade of *Saccharomyceteae*. Yeasts in this clade are primarily environmental organisms isolated from niches overlapping those of *S. cerevisiae*. Some of these species are known for contaminating industrial fermentations, such as the acid-tolerant species *Zygosaccharomyces bailii*; however others are used for industrial purposes, such as *Zygosaccharomyces rouxii* in the production of soy-sauce[10]. This clade has recently garnered attention because of the observation of horizontal gene transfer of large fragments of DNA from *Z. bailii* and *T. microellipsoides* to the genome of commercial wine strains[11], which endows these strains with superior nitrogen utilization properties during fermentations[12].

In studying *T. microellipsoides* for un-related reasons, we serendipitously discovered that this species undergoes a phenotypic switch reminiscent of the white-to-opaque switch documented exclusively in *C. albicans*, *C. tropicalis*, and *C. dubliniensis*[5,13,14]. This was surprising given that *T. microellipsoides* and these species are hundreds of millions of years divergent. Here, we present a characterization of this phenotypic switch in *T. microellipsoides*, wherein typical small white colonies switch to larger, darker and flatter colonies with an opaque texture and enlarged cells. We found that the switching phenotype was unique compared to the *C. albicans* white-to-opaque transition in terms of transcriptome profiles and cellular DNA content. We further characterized each

cell type and found that the two different morphologies of *T. microellipsoides* have different properties in terms of growth rate, stress tolerance, and genetic stability. The most important aspect of our observations is that phenotypic switching between highly divergent yeast-cell types exists in at least one species in the ZT clade, suggesting that more yeasts than previously thought may undergo dramatic morphological transitions resembling the white-to-opaque switch.

## Results

### *T. microellipsoides* undergoes a white-to-opaque-like morphological switch.
In working with the yeast *T. microellipsoides*, we serendipitously discovered that the species undergoes a morphology switch on agar plates that is reminiscent of the white-to-opaque switch observed in *C. albicans*. *T. microellipsoides* (NCYC 2568) typically exists as small ellipsoid budding yeast that forms shiny white colonies on agar plates, with a colony texture similar to that of *S. cerevisiae* or white *C. albicans* (Fig. 1a, b). After 12 days on YEG agar plates at room temperature, we observed that this yeast undergoes a morphological switch, wherein a white yeast colony spontaneously gives rise to a darker dull-textured 'opaque' sector that can be stably propagated (Fig. 1a). Because of the similarity in appearance to *C. albicans* white-to-opaque switching, we denote the cells from the white colony as 'white' cells and from the sector as 'opaque' cells. Microscopically, *T. microellipsoides* opaque cells appear as larger, slightly ellipsoid budding yeast (Fig. 1b), which is distinct from *C. albicans* opaque cells, which are very elongated compared to white cells. As increased cell size is often associated with increased ploidy, we measured the DNA content of white and opaque cells by flow cytometry, and observed a two- to fourfold increase in the DNA content of opaque cells, as compared to white cells (Fig. 1d, Supplementary Figs. 1 and 2). This increase in DNA content distinguishes the *T. microellipsoides* morphology switch from that of *C. albicans* white-to-opaque switching, which does not alter ploidy.

### Measurements of white-to-opaque and opaque-to-white switching rates.
We measured the frequency of white-to-opaque switching in *T. microellipsoides* by plating white cells for single colonies on YEG, growing for 14 days, and counting the rate of colony sectoring to opaque. We found that the white-to-opaque switch frequency was approximately 0.13% under these conditions (Fig. 1c). White-to-opaque switching in *C. albicans* is increased in several different conditions, including growth on *N*-acetylglucosamine (GlNAc), 5% $CO_2$, and by UV irradiation[5,15,16]. To determine if the switching frequency of *T. microellipsoides* could be increased, we analysed the switching frequency on GlcNAc and after UV exposure. While GlcNAc containing media had no effect (Supplementary Fig. 3), we observed an approximately 300-fold induction of white-to-opaque switching after UV irradiation (Fig. 1c).

To test for opaque-to-white switching, we plated opaque cells for single colonies, and observed that switching in this direction occurs very differently than white-to-opaque switching. After 14 days on plates, *T. microellipsoides* opaque colonies formed lighter-coloured protrusions, which were similar in texture to white cell colonies, on the surface of 100% of colonies (Fig. 1e). In *C. albicans* and other related *Candida* species that switch, opaque-to-white switching occurs similarly to white-to-opaque, wherein a colony will form a sector back to the other morphotype, or less commonly a full colony will switch back[17]. We therefore evaluated whether these protrusions represented a genuine reverse switch by re-streaking them onto fresh YEG plates, and then evaluating their phenotype and DNA content.

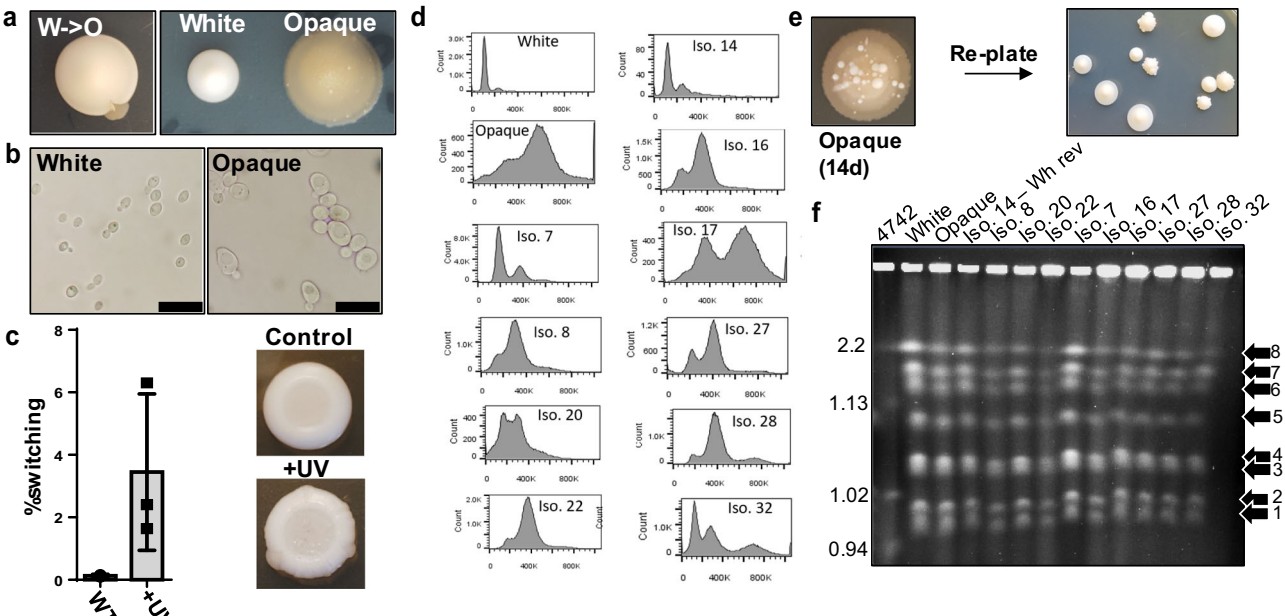

**Fig. 1 Measurements of white-to-opaque and opaque-to-white switching and DNA content analysis in *T. microellipsoides*. a** Example images of white-to-opaque sector-based switching (left) and adjacent white and opaque colonies (right). **b** Phase-contract microscopy images of white and opaque cells at ×1000 magnification. **c** Quantification of white-to-opaque switching under standard conditions and after UV-treatment; $N = 3$ independent experiments. An example of untreated and UV-treated cells are shown on the right, with opaque sectors denoted by arrows. **d** DNA content analysis of white, opaque, and ten opaque cell derivatives (including a white cell revertant, Iso 14), as measured by flow cytometry. **e** Example image of reverse switching of an opaque colony (left), and colonies derived from re-plating the same colony (right). **f** CHEF gel analysis of chromosome banding pattern of white, opaque, and ten opaque cell derivatives (including a white cell revertant, Iso 14). *S. cerevisiae* BY4742 denotes size standards. Error bars represent the standard deviation of the mean; significance values are shown in Supplementary Data 1.

We found that approximately 10% of individual colonies isolated from these structures were genuine white cells, both phenotypically (based on cell size and colony texture) and by DNA content (Fig. 1d). This indicates that a genuine reversal of the opaque-to-white phenotype does occur, although it is difficult to assign a number to reverse switching of this nature. We also observed a large degree of phenotypic heterogeneity in the colony appearance and growth rate of re-streaks from protrusions that did not appear to be white cells (i.e. the other 90%). A number of these isolates had DNA content profiles intermediate between white and opaque cells, and several had either associated growth defects, or heterogeneous colony morphologies with multiple sectoring regions (Fig. 1e, Supplementary Fig. 2). Several of these features are indicative of aneuploidy (i.e. growth defects, DNA content, and sectoring[18,21,22]), and because these colony isolates arose from higher ploidy opaque colonies, we hypothesized that this phenotype might be symptomatic of chromosome loss, similar to what occurs in mated *C. albicans* tetraploid cells[19]. We attempted to observe possible chromosomal loss and/or rearrangements via CHEF gel analysis (karyotyping) of white, opaque, and 10 diverse opaque-derived isolates, including a genuine white cell revertant. Consistent with previous studies, our data show that *T. microellipsoides* contains eight chromosomes, all between the sizes of 0.94 and 2.2 Mb (Fig. 1f). Importantly, however, across all samples there were no conspicuous differences in chromosomal banding patterns or brightness (Fig. 1f). Thus, at the karyotype level, we did not observe any evidence of gross chromosomal loss or rearrangements during opaque-to-white switching.

**Genome sequencing of white and opaque cells**. Having established that *T. microellipsoides* undergoes a phenotypic switch from white-to-opaque, that this switch involves a ploidy increase,

and that the switch appears to be reversible, but is also accompanied by a burst in genotypic and phenotypic heterogeneity in the population, we next wanted to better characterize genotypic changes in switching yeast.

We performed whole-genome sequencing of white and opaque cells to determine if a genetic change could provide an explanatory mechanism for white-to-opaque switching. The results of this sequencing are depicted in a Circos plot[20] (Fig. 2), which shows an alignment of white and opaque genome scaffolds; this is a graphical representation of the mapping of scaffolds/contigs larger than 1 kb for both assemblies to one another. We show the best match for sequences longer than 1 kb with an identity of at least 70%, which is stringent enough to look at major rearrangements only. Scaffolds generally mapped to their corresponding counterparts; however, possible translocation events were identified between scaffold 2 white and scaffold 6 opaque (length = 19.2 kb, id = 100%), scaffold 3 white and scaffold 5 opaque (length = 1.3 kb, id = 93%), and scaffold 3 white and scaffold 5 opaque (length = 2.7 kb, id = 77%). Importantly, our analysis of the annotated genes within identified translocated regions did not indicate the presence of any gene(s) that were obviously explanatory of the phenotypic switch between white and opaque cells (Supplementary Table 1, Supplementary Fig. 4). Taken together, our analyses suggest that the mechanism of forward switching is either not a genetic change or, alternatively, one that is not like any that have been previously observed.

**mRNA-sequencing analysis of white and opaque cells**. In the absence of obvious explanatory genetic changes between white and opaque cells, we opted to characterize the transcriptome of both cell types to gain a deeper understanding of what functional

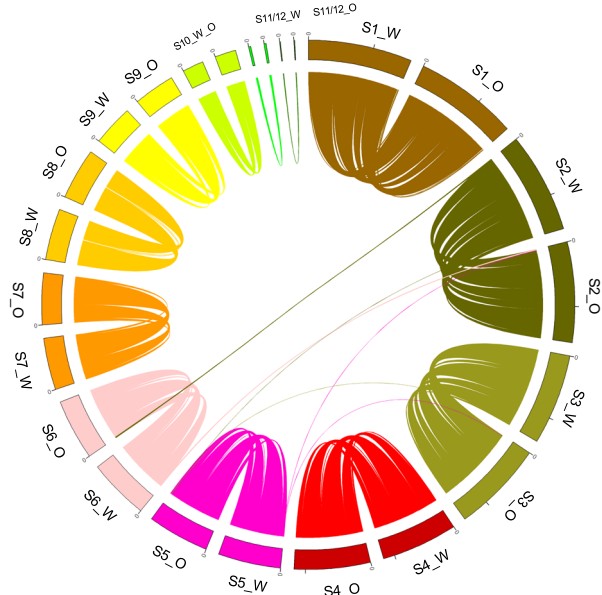

**Fig. 2 Genomic comparison of white and opaque cells shown in a Circos plot, comparing assemblies of *T. microellipsoides* white and opaque cells at the scaffold level.** White and opaque cell scaffolds are represented by the letters S#_"W" and "O" (Scaffold Number White/Opaque), respectively. Links represent matching sequences of at least 1 kb in length and 70% sequence identity and are coloured based on their scaffold of origin. Most scaffolds sequences are similar between assemblies. Possible translocation events are represented by pairs of links, more precisely from chr w 2 to chr o VI, chr w 2 to chr o V, chr w 5 to chr o III, and their respective counterparts.

differences may exist, and to attempt to explain the mechanism(s) of morphology switching. To this end, we compared the transcriptome of matched log-phase cultures of white and opaque cells grown in YEG media. Overall, we identified 4465 transcripts in both white and opaque cells, representing coverage of approximately 85% of the open-reading frames predicted by a reference genome of CLIB830T[21]. Between white and opaque cells, 511 transcripts (11.4% of total transcript number) were differentially regulated at least fourfold, with 249 transcripts more highly expressed in opaque cells and 262 transcripts more highly expressed in white cells (Supplementary Data 1). Gene ontology (GO)-term analysis indicates that transcripts enriched in opaque cells were strongly associated with terms such as 'ribosome biogenesis' ($p$ value 2.54e-47), ribonucleoprotein complex biogenesis ($p$ value 8.48e-42), and many other rRNA associated processes, whereas white-enriched transcripts were associated with the term 'carbohydrate metabolic process' ($p$ value 0.00145). When grouped by general gene function, such as metabolic function or transcription, we observed that white cells have a relatively diverse pattern of enriched gene transcripts (Fig. 3a), whereas the majority of opaque-enriched transcripts were associated with ribosome biogenesis (Fig. 3b). STRING analysis[22], which creates groupings based on connections between known and predicted protein–protein interactions, showed a single hub of putative interacting genes in opaque-enriched transcripts, again strongly associated with ribosome biogenesis (Supplementary Fig. 5). Analysis of white-enriched transcripts revealed two related clusters, primarily involved in non-fermentable carbohydrate source utilization/post-diauxic shift growth and general stress response (Supplementary Fig. 6).

The white-to-opaque switch in *C. albicans* is governed by a well-characterized transcription factor (TF) network, including *WOR1*, *EFG1*, and others[23]. Thus, we endeavoured to search for

differentially regulated TFs in white and opaque cells of *T. microellipsoides* that could help explain the mechanism of switching. In opaque cells, we found that three TFs were more highly expressed, two of which are homologous to *BAS1*, which is involved in regulating expression of genes involved in purine and histidine biosynthesis[24], and one of which is homologous to Mnn4, a regulator of cell-wall biogenesis[25]. In white cells, homologues of 14 TFs were more highly expressed compared to opaque cells, including *ASG1*, *XBP1*, *ACE2*, *CAT8*, *ADR1*, *PDR1*, *RDR1*, *YRM1*, *IME1*, *RPI1*, and several others of unknown function (Supplementary Data 1). Notably, many of these are directly involved in regulation of stress resistance or are induced during the diauxic shift in *S. cerevisiae*[26,27]; however, none stand out as obvious putative TFs responsible for morphological switching.

Overall, while there were changes in the levels of global transcription regulators (TR), neither the *WOR1* or *EFG1* homologue (the central regulators of switching in *C. albicans*)[5] nor homologues of any other *C. albicans* white-to-opaque regulators were differentially regulated between the two cell types, suggesting these proteins are not involved in this phenotypic switch in *T. microellipsoides*. For additional confirmation, we created a deletion of the *EFG1* homologue in *T. microellipsoides* and evaluated white-to-opaque switching. In *C. albicans*, *EFG1* deletion results in a high rate of switching to the opaque phase[17,28], however we did not observe any effect on *T. microellipsoides* white-to-opaque switching (Supplementary Fig. 7).

While we observed relatively little similarity between the transcriptional profile of *T. microellipsoides* and *C. albicans* white and opaque cells, we did make several general observations about these two cell types in *T. microellipsoides*, which allowed us to hypothesize about possible specialized functions of each cell morphology. Firstly, compared to white cells, opaque cells seem to be primed for proliferation based on the whole-scale increase in expression of genes involved in ribosome biogenesis and tRNA synthesis[29], as well as increased expression of the nitrogen uptake genes *GAP1* and *MEP2* (verified by RT-qPCR), which are under nitrogen catabolite repression in other species (Supplementary Data 1, Supplementary Fig. 8)[30]. Secondly, higher expression levels of enzymes involved in glycogen and trehalose synthesis, metabolism of complex carbon sources, and catalases[27] suggest that, compared to opaque cells, white cells may be primed to handle stress conditions, such as nutrient starvation in stationary phase, growth on alternate carbon sources, and conditions of oxidative stress and/or heat shock. Thus, we posited that white cells would tolerate exposure to these stressors more effectively than opaque cells, whereas opaque cells would be more sensitive to these stressors, but would proliferate at a higher rate under optimal, non-stressful conditions.

**Sensitivity of white and opaque cells to nutrient, oxidative, stationary phase, and other cellular stressors.** We endeavoured to characterize the two cell types by comparing the growth of white and opaque cells on YEG or YEG containing 3 mM $H_2O_2$, YNB containing various complex carbon or nitrogen sources, and exposure of both cell types of additional commonly encountered environmental stressors. In doing so, we first observed that there were no obvious differences in growth for white cells compared to opaque on any carbon or nitrogen sources tested, indicating that neither cell type had an observable growth advantage in these conditions (Supplementary Fig. 9). Additionally, we observed that opaque cells form notably larger colonies than white cells on YEG agar plates, consistent with an increase in cell size and cell-mass proliferation. To quantify this, we measured the wet mass of white and opaque colonies, and found that the mass of opaque colonies was approximately 2.5-fold greater than white colonies (Supplementary

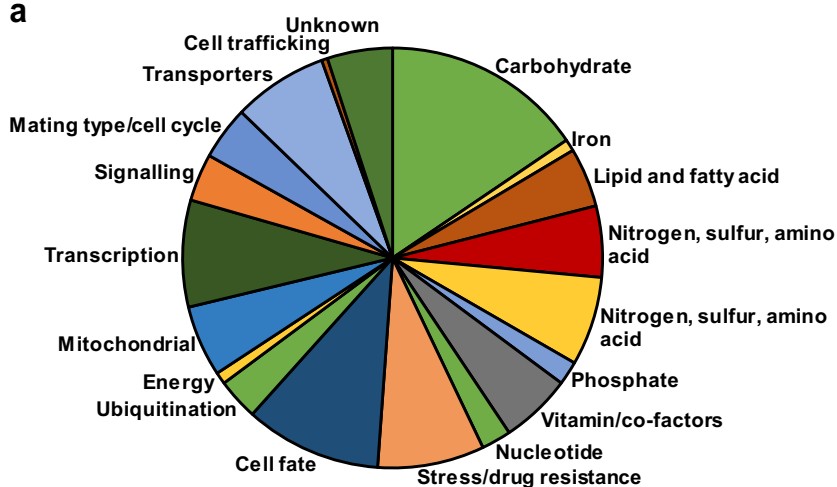

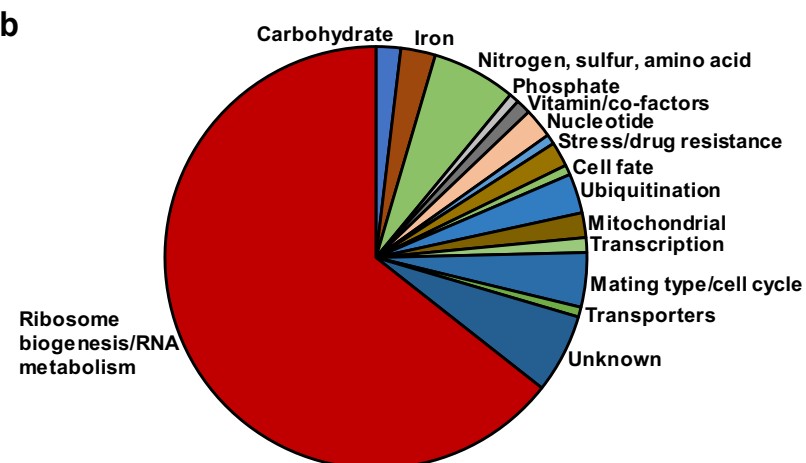

**Fig. 3 Gene expression analysis in white and opaque cells as measured by mRNA-seq. a** Pie chart representation of gene transcripts enriched in white cells compared to opaque cells, grouped by metabolism, ribosome biogenesis, transcription, and other major categories. **b** Pie chart representation of genes transcripts enriched in opaque cells compared to white cells, grouped as in panel **a**.

Fig. 10A), consistent with our gross observations; we also measured the number of cells per colony and observed that white colonies have approximately threefold more cells per colony than opaque colonies, reflecting the large difference in cell size (Supplementary Fig. 10B).

Next, we tested the sensitivity of white and opaque cells to oxidative stress by exposure to $H_2O_2$. Consistent with our prediction based on the mRNA-seq analyses, we found that opaque cell survival was greatly reduced compared to white cells when grown in the presence of 3 mM $H_2O_2$ (Fig. 4a), indicating that they are indeed more sensitive to oxidative stress.

We also tested tolerance of the two cell types to starvation conditions encountered in stationary-phase cultures. This experiment was performed by inoculating white and opaque cells in liquid YEG medium and monitoring their growth over time by OD measurements and viability by methylene blue-staining and flow cytometry. We observed that white cells grow to a much higher density in the stationary phase; however, the wet weight per ml of white and opaque cells in the stationary-phase cultures were similar. Thus, these differences in growth may reflect the difference in cell size between white and opaque cells, as noted in the previous section (Fig. 4b–d). Viability measurements, however, showed a striking difference between white and opaque cells. Even during early growth timepoints (i.e. log phase), the

viability of opaque cells was never greater than 95%, and over time, viability decreased dramatically such that after 5 days in culture opaque cells were only 30% viable and continued to slowly decrease, whereas white cells remained at least 91% viable for the entire time course (Fig. 4e). Additionally, we tested the viability of white and opaque cells on agar plates after 14 days of growth and observed a similar viability reduction in opaque cells compared to white cells, both on YEG and more nutrient-rich YEPD medium (Fig. 4f).

To further expand on differences between the two cell morphologies, we explored growth of white and opaque cells under a number of different stressful conditions that this species may encounter, including exposure to desiccation, cell-wall degradation (with the enzyme zymolyase), heat shock, the cell-wall stressors calcofluor white and Congo red, and growth at different temperatures.

We found that after desiccation for 24 h, approximately 5% of white cells survive, whereas there was no detectable survival of opaque cells (Fig. 4g). In testing additional cell-wall stressors, we found that opaque and white cells have similar growth rates on plates containing calcofluor white and Congo red (Supplementary Fig. 11). In testing for sensitivity of both cell types to zymolyase, we found that 100% of white cells survived treatment with this

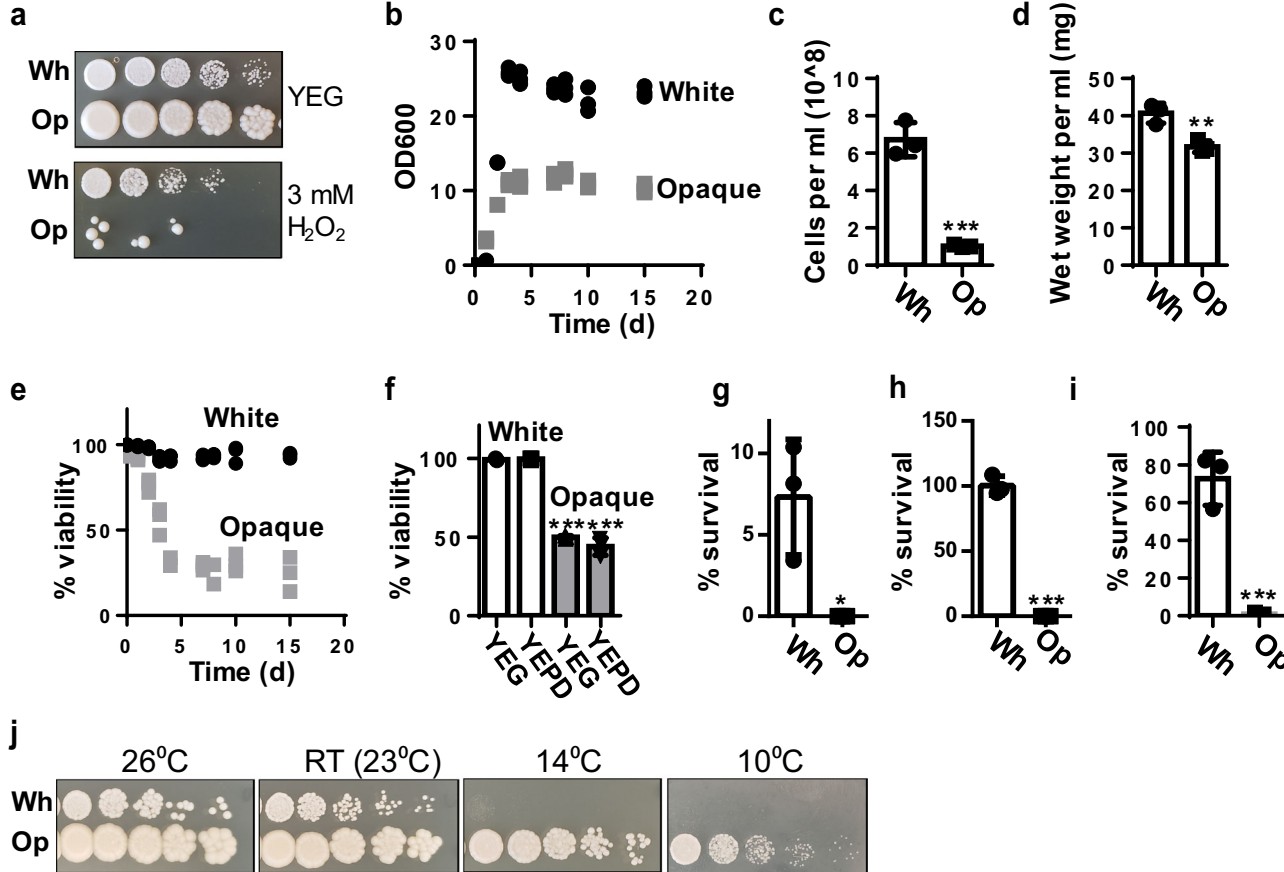

**Fig. 4 Comparison of white and opaque cell resistance to oxidative, stationary phase, and other stressors. a** Dilution series of white and opaque cells on YEG and YEG + 3 mM $H_2O_2$. **b** Growth curves of white and opaque cells grown in liquid YEG medium. $N = 3$ independent cultures for panels **b–e**. **c** Measurements of cell number per millilitre in the stationary phase of growth (day 7) for white and opaque cells. **d** Wet weight per millilitre of stationary-phase cultures of white and opaque cells (day 7). **e** Viability measurements over time of white and opaque liquid cultures. **f** Viability of white and opaque colonies after 14 days of growth on either YEG or YEPD. $N = 3$ independent colonies. **g** Survival of white and opaque cells after 24 h of desiccation. $N = 3$ independent experiments for panels **g–i**. **h** Survival of white and opaque cells after digestion with zymolyase for 2 h. **i** Survival of white and opaque cells after 5 min of heat shock at 50 °C. **j** Dilution series comparing the growth rate of white and opaque cells at 26 °C, 23 °C (RT), 14 °C, and 10 °C. Error bars represent the standard deviation of the mean; significance values are shown in Supplementary Data 1.

enzyme, whereas surviving opaque cells were not detected (Fig. 4h). After heat-shock treatment for 2 min at 50 °C, the survival rate of white cells was approximately 70% compared to <1% for opaque cells (Fig. 4i).

Lastly, we tested growth of white and opaque cells at different temperatures, and observed that at 23 °C (room temperature) and 26 °C, white and opaque cells grew similarly, however at lower temperatures, such at 14 °C and, more strikingly, at 10 °C, opaque cells continued to grow well, whereas white cells grew at a highly reduced rate (Fig. 4j). To assess if the superior low-temperature growth rate of opaque cells translated into a competitive advantage, we performed direct competition assays between white and opaque cells in three different conditions: (1) growth on a plate at a room temperature (2) growth on a plate at 10 °C, and (3) growth in liquid culture at room temperature. We observed that after 48 and 96 h, opaque cells showed a modest competitive advantage compared to white cells when grown at room temperature. When grown at 10 °C, the growth advantage of opaque cells was much greater, with opaque cells representing a majority of the population after 96 h. However, in liquid culture at room temperature, white cells show a competitive advantage over opaque cells after 96 h (Supplementary Fig. 12).

Overall, our results indicate that white cells are more resistant to multiple physiological stressors (i.e. oxidative and stationary-phase stress, heat shock, etc.), as compared to opaque cells, which is consistent with several of our predictions based on the transcriptional profiles of the two cell types. Furthermore, while opaque cells generate more cell mass per colony on agar plates and grow well at colder temperatures, they quickly lose viability over time in both solid and liquid based growth conditions.

**Expression of oxidative and stationary-phase stress-response genes in white and opaque cells.** In the previous section, we established that there are innate differences in stress tolerance between white and opaque cells. Such differences may indicate some cellular specializations of each cell type as a result of transcriptional re-wiring of stress-response pathways. To test if this was the case in opaque cells, we examined the transcriptional induction of homologues of canonical oxidative stress and general stress-response genes in response to hydrogen peroxide. In this experiment we observed that upon exposure to 3 mM $H_2O_2$ in liquid culture, white cells grew at 75% the rate of untreated cells, whereas opaque cells grew at 14.2% the rate, reflecting the

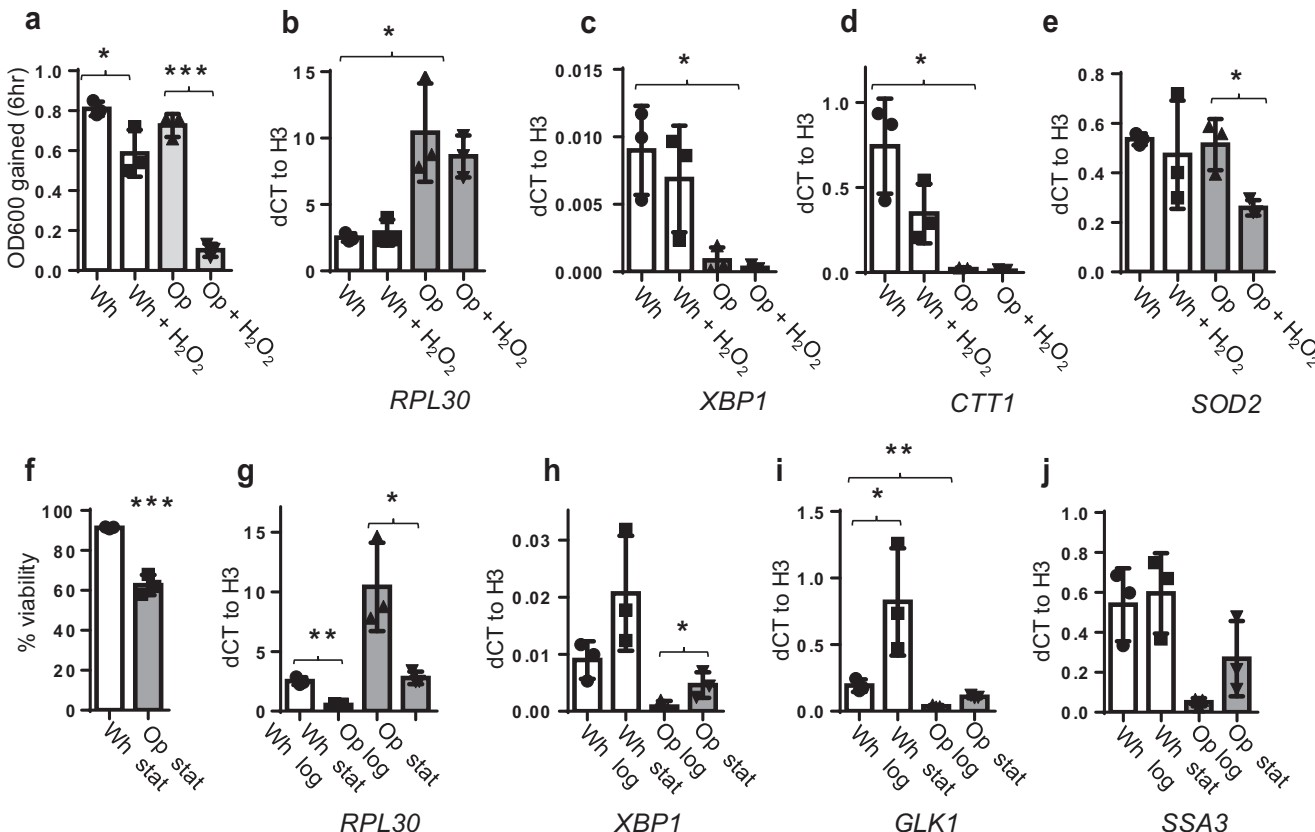

**Fig. 5 Comparison of white and opaque cell transcriptional responses under oxidative and stationary-phase stress. a** $OD_{600}$ measurement of white and opaque cells after 6 h of growth with or without $H_2O_2$ treatment. **b–e** Relative expression of *RPL30* (**b**), *XBP1* (**c**), *CTT1* (**d**), and *SOD2* (**e**) in white, opaque, and both cell types grown with $H_2O_2$, as measured by RT-qPCR. **f** Viability of white and opaque stationary-phase liquid cultures at time of harvest. **g–j** Relative expression of *RPL30* (**g**), *XBP1* (**h**), *GLK1* (**i**), and *SSA3* (**j**) in white and opaque cells in the stationary phase, as measured by RT-qPCR. $N = 3$ independent cultures for all panels. Error bars represent the standard deviation of the mean; significance values are shown in Supplementary Data 1.

sensitivities observed by dilution series (Fig. 5a). For these experiments, we opted to study the expression of four genes: homologues of two canonical oxidative stress-response genes, Catalase T (*CTT1*)[31] and superoxide dismutase 2 (*SOD2*)[32], the homologue of the stress-associated transcriptional regulator *Xho*I site-binding protein 1 (*XBP1*)[33], and a homologue of the ribosome biosynthesis-associated gene Ribosomal Protein of the Large subunit 30 (*RPL30*)[34]. These genes were chosen based on homology to *S. cerevisiae* proteins, and in the case of *CTT1*, *RPL30*, and *XBP1*, the differential expression profiles in our RNAseq experiment (Supplemental Data 1). Our expression analyses indicated that none of these genes are induced in response to $H_2O_2$ treatment in either white or opaque cells. Specifically, we found that both *CTT1* and *XRN1* are more highly expressed in white cells compared to opaque cells regardless of $H_2O_2$ treatment, *SOD2* is not different between the two cell types (Fig. 5b–e), and *RPL30* is more highly expressed in opaque cells; all of these results were also consistent with our mRNA-seq observations (Supplementary Data 1). Thus, we conclude that homologues of key genes involved in the oxidative stress response are not induced in *T. microellipsoides* in response to this stressor, and instead they appear to be differentially regulated by cell type, which is reflective of their resistance to this stressor.

To test for induction of genes specific for stationary-phase survival, we grew cultures of white and opaque cells for 3 days and harvested samples for RT-qPCR analysis; at this timepoint white cells remained between 95 and 100% viable, whereas opaque cells

had dropped to ~60% viability (Fig. 5f). We examined expression of homologues of *XBP1*, Glucokinase 1 (*GLK1*)[35], Stress-Seventy subfamily A 3 (*SSA3*)[36], and *RPL30*[34], as all of these genes are known to be differentially regulated in log versus stationary phase growth in *S. cerevisiae*. Furthermore, *GLK1*, *XBP1*, and *SSA3* play key roles in *S. cerevisiae* stationary-phase survival[26,33], and were differentially expressed between white and opaque cells in our mRNA-seq analyses (Supplementary Data 1). In contrast to the oxidative stress response, a moderate induction of *XBP1* and *GLK1* was detected in stationary-phase cells of both types, as compared to log-phase cells; *SSA3* was not differentially expressed (Fig. 5g–j). The opposite trend was true with regards to *RPL30* expression; levels were reduced in stationary phase compared to log phase in both white and opaque cells. In addition, there were generalized differences in expression of *XBP1*, *GLK1*, *SSA3*, and *RPL30* between white and opaque cells (Fig. 5g–j). Although *XBP1* and *GLK1* exhibit a stationary-phase induction in opaque cells, and the induction is of a similar magnitude to white cells (~4-fold), their overall levels were 5 to 8-fold lower than in white cells, regardless of induction. The opposite was also true for *RPL30*, where there was a decrease in transcription in stationary phase compared to log phase in both white and opaque cells, however overall levels of this transcript were ~5-fold higher in opaque vs. white cells. The pattern we observed from these data is that white and opaque cells differ substantially in their innate expression of these marker genes. Within each cell type, expression can be induced by exposure to the stationary-phase stressor, however the levels of induction are of a

much lower magnitude than the innate differences in expression level between the cell types. Taken together, these transcriptional and phenotypic data provide evidence that white and opaque cells may be re-wired in terms of oxidative stress-response and growth-phase related gene induction, and these differences could reflect potential specializations of each cell type.

## Discussion

Here, we report the discovery of a phenotypic switch in the ZT-clade species *T. microellipsoides* from small white budding yeast to a larger budding yeast with an opaque colony texture. Prior to our work, this phenomenon had only been observed in *C. albicans* and two other related species, all members of the CTG clade of *Saccharomycetacea*. As such, our findings suggest that phenotypic switching from one yeast type to another in ascomycetous fungi, specifically white-to-opaque-like switching, may be more common that previously thought, and that study of un-characterized yeasts may reveal additional species that possess this capability.

White-to-opaque switching in *T. microellipsoides* is morphologically reminiscent of white-to-opaque switching in the three canonical *Candida* species. This may suggest that morphology switching of this fashion has either independently evolved in phylogenetically distant yeast species or is an inherent ancestral trait in ascomycetous yeast in which cellular functions of the alternate (opaque) cell type diverged. The white-to-opaque switch in *C. albicans* is one of the most well studied epigenetic switches in nature[5], and significant understanding of fundamental phenomena such as the interaction and function of complex transcription factor loops in eukaryotic cells have been gained from its study. Further characterization of the white-to-opaque switch in *T. microellipsoides* may provide similar insights into diverse biological processes.

Compared to our basic definitions of *T. microellipsoides* white and opaque cells, the two cell types in *C. albicans* have been defined in detail over many years of study. In *C. albicans*, opaque cells display multiple functions compared to white cells, including specialization for mating and colonization of different host tissues, immune evasion in certain situations, and a metabolism specific for growth in a limited number of conditions[5–7,16]. White cells are thought to be the generalist cell type, with growth advantages compared to opaque cells under most conditions, and they are considered the 'basal state' of *C. albicans*[37]. In this sense, morphology switching in *T. microellipsoides* and *C. albicans* is directly relatable, as white cells also appear to be the 'basal' state in *T. microellipsoides* and act as metabolic and stress-tolerance generalists, whereas opaque cells exist in a specialized metabolic state (i.e. higher expression of ribosome biogenesis genes) and only appear to be superior to white cells in a specific subset of conditions. A high throughput Biolog-based comparison of the two cell types, as was done in *C. albicans*[37], would provide valuable insight into the full phenotypic profile of the two cell types in *T. microellipsoides*.

We also note a potential similarity of *T. microellipsoides* white cells and *S. cerevisiae* spores in addition to the fact they are both haploid. In *S. cerevisiae* (and most yeasts), zymolyase degrades the cell wall of vegetative cells, and the ascus surrounding spores, but not spores themselves[38]. This property of *S. cerevisiae* spores is critical for their survival passaging through *Drosophila melanogaster* (fruit fly) digestive tracts[39]. In natural environments, *S. cerevisiae* are often consumed by *D. melanogaster*; and surviving passage through the GI tract of this and related organisms allows the spread of the yeast to new niches[39]. Our results demonstrate that *T. microellipsoides* white cells are insensitive to zymolyase, which may also allow the cells to passage through insect species. These two forms may represent different evolutionary mechanisms

of handing cell-wall stressors, or perhaps a similar evolutionary origin, as a *Torulaspora* species is thought to be one of the two yeasts which hybridized to form the whole-genome duplication clade of yeasts[9].

Based on our observations herein, we propose the following model for the white-to-opaque morphology switch in *T. microellipsoides*: white and opaque cells represent two distinct phases of this species, which function optimally under different specialized conditions. The basal state of *T. microellipsoides* is likely the white phase, where cells grow slowly and conservatively, and are highly resistant to many environmental stressors; being an environmental organism, it is likely these represent the most commonly encountered conditions for this species. Switching from white-to-opaque is generally stochastic; however, certain stimuli, such as UV exposure, increase the switching rate. Once formed, opaque cells represent a specialized state of *T. microellipsoides* that is optimized for rapid growth and utilization of nutrients, as well as growth at low temperatures. When conditions are optimal (i.e. rich in resources), they function to rapidly expand the population to fill the available niche quickly. When resources are exhausted, the population of opaque cells collapses, undergoing a high amount of cell death, as well as generating progeny with diverse phenotypes (possibly via chromosome loss or rearrangements), which provide diverse starting material for either occupation of a new niche, or for long-term survival of stress, as a notable percentage of these progeny are genuine white cell revertants.

Overall, the most important implication of our findings is that morphology switching resembling white-to-opaque switching occurs in the ZT-clade yeast *T. microellipsoides*. Importantly, *T. microellipsoides* is hundreds of millions years diverged from *C. albicans* and other yeasts that undergo morphologic switching of a similar nature, thus this trait may be more common than previously thought and, therefore, is an point of interest for future study. Indeed, there are many avenues of further investigation, and our initial characterization provides several tools, such as the genomic and mRNA-seq data for white and opaque cells as well as methods of genetic manipulation in this species, that facilitate these future studies.

## Methods

**Strains, media, and growth conditions.** *Escherichia coli* strain DH5α was used for cloning and plasmid propagation. *T. microellipsoides* strain NCYC 2568 was used in all studies. Unless otherwise indicated, yeast cells were propagated on yeast extract glucose [1% yeast extract, 2% dextrose] at 23 °C for all assays. For colony morphology assays, cells were propagated for 14 days at 23 °C on YEG agar plates. For $H_2O_2$ sensitivity assays, cells were grown on 0.3 or 3 mM $H_2O_2$ containing YEG agar plates for 5 days. In liquid culture, cells were grown in YEG medium in test tubes on a roller drum at 23 °C for the times indicated with the exception of the mRNA-seq analysis, where triplicate cultures of white and opaque cells were grown in 50 ml cultures in a 250 ml Erlenmeyer flask at 250 r.p.m. at 23 °C. For chemical transformations, cells were plated on geneticin (G418) at a concentration of 200 μg/ml. All strains are given in Supplementary Table 2.

**Genome sequencing and assembly.** Genomic DNA was purified using a standard phenol–chloroform-based extraction. DNA was then treated with RNaseA for 1 h at 37 °C, and re-purified with an additional phenol–chloroform extraction. Sequencing libraries were prepared using the Nextera DNA Flex library preparation kit, according to the manufacturer's instructions (Illumina). Libraries were pooled and loaded onto a NextSeq High Output flow cell, generating paired-end 150 bp reads. Raw base call data (bcl) were converted into FastQ format using the bcl2fastq conversion software from Illumina (version 2.19). Library preparation and sequencing were performed at Sequencing and Bioinformatics Consortium at the University of British Columbia (Vancouver, Canada). Quality Control was performed using FASTQC[40]. Reads were aligned against the *T. microellipsoides* reference genome (genbank ID CLIB830T)[21] with 100% of reads aligned to the reference genome for both libraries and an average sequencing depth of 123 and 91 for opaque and white samples, respectively. Reads for white and opaque samples were also assembled into scaffolds using Abyss[41], with N50 of 151741 and 187462 for white and opaque samples, respectively. Pairwise comparison of the assembled scaffolds was done using LASTZ[42] and represented within a Circos plot[20].

All sequencing data have been deposited on the Gene Expression Omnibus served (NCBI), reference number GSE136991.

**mRNA-sequencing and analysis**. Total RNA was purified using a standard hot acid phenol–chloroform extraction[43]. RNA was then treated with DNaseI for 1 h at 37 °C, and re-purified with an additional acid phenol–chloroform extraction. RNA was quantified using Qubit fluorometry and 500 ng of each sample was used to prepare sequencing libraries with the TruSeq Stranded mRNA library preparation kit, according to the manufacturer's instructions (Illumina). Libraries were pooled and loaded onto a NextSeq High Output flow cell, generating paired-end 75 bp reads. Raw base call data (bcl) were converted into FastQ format using the bcl2fastq conversion software from Illumina (version 2.19). Library preparation and sequencing were performed at Sequencing and Bioinformatics Consortium at the University of British Columbia (Vancouver, Canada). Quality Control was performed using FASTQC[40]. RNAseq reads aligned to the CLIB830T reference genome using STAR[44] (see Supplementary Data 1 for sequence depth and map ability). Transcriptomes for all libraries were assembled and merged using cufflinks[45]. Annotations for transcript sequences were derived from the Entrez [NCBI resource coordinator] protein sequence database for *S. cerevisiae* using BLAST[46]. Best matches (median length of match = 464 bp, median percent identity = 62) were retrieved, and defined as transcript annotations into an ensembl gff3 file [https://uswest.ensembl.org/info/website/upload/gff3.html]. The latter was then used to quantify transcripts levels in all analysed samples. Differential expression was assessed using DEseq 2.0[47]. All sequencing data have been deposited on the Gene Expression Omnibus served (NCBI), accession number GSE136991.

**White-to-opaque/opaque-to-white switching assays**. White-to-opaque switching was quantified by plating cells for ~100 CFU per plate on YEG agar and incubating for 14 days at 23 °C. Colonies were then visually inspected for opaque sectors, which are easily identified based on the texture and colour difference of the two cell types.

**Pulse-field gel electrophoresis**. White, opaque, and opaque-derived cells were grown in YEG overnight at room temperature on a roller drum. Cell density for each culture was measured by hemocytometer counting, and $1 \times 10^8$ cells of each culture were harvested and washed once in 1 ml of dH$_2$O. Cells were then washed twice in 500 µl of 0.5 M EDTA. Cells were then embedded in 0.5% Agarose plugs (made with 0.5 M EDTA). Plugs were then washed once in 500 µl of 0.5 M EDTA, and the supernatant removed. Five microlitres of a solution containing 5 mg/ml Proteinase K, 1% *N*-lauroyl sarcosine, and 0.5 M EDTA was added to each individual plug and incubated at 50 °C for 24 h. Plugs were then washed twice with 500 µl of 0.5 M EDTA and stored at 4 °C. Plugs were then loaded into a 0.8% agarose/TBE gel, and chromosomes separated on a contour-clamped homogeneous-electric-field gel electrophoresis (CHEF) gel apparatus in 0.5× TBE [45 mM Tris-borate pH 8.0, 1 mM EDTA] at 75 V cooled to 14 °C. Pulse times were 120 s for 48 h, followed by 60 s for 24 h. Gels were then stained with ethidium bromide (0.5 µg/ml) and visualized on an ultra-violet transilluminator.

**Competitive fitness assays**. White and opaque cells were grown to log phase (OD$_{600}$ of 0.5) in YEG media. One millilitre of each cell type was harvested by centrifugation, washed once in dH$_2$O, and re-suspended in 500 µl of dH$_2$O. Cells per millilitre were measured by hemocytometer counts for each suspension, and $1 \times 10^6$ cells of each cell type were mixed in a separate tube and vortexed to mix well. Serial dilutions of this mixture were plated to determine the starting relative ratio of white and opaque colonies, which was approximately 50% of each. For test conditions, 5 µl of this mixture was spotted onto agar plates and incubated at either room temperature (23 °C) or 10 °C for 48 or 96 h, after which a sample was collected, re-suspended in 500 µl of H$_2$O, and serial dilutions were plated to determine the relative CFU of white and opaque cells. For the liquid culture experiment, 5 µl of the initial mixture was inoculated into 5 ml of fresh YEG media and incubated on a roller drum at room temperature for 96 h. To calculate the recovered/inoculum ratio, CFU of each cell type recovered in each specific experimental condition was divided by the initial inoculum CFU ratio.

**Zymolase sensitivity**. White and opaque cells were grown to the log phase (OD$_{600}$ of 0.5) in YEG media. Two hundred microlitres of culture were harvested, and washed once in 1 ml of dH$_2$O and pelleted. The pellet was re-suspended in 100 µl of Zymolyase 100-T (units/ml), and incubated at 30 °C for 1 h. One millilitre of water was then added, and the suspension mixed by repeated inversions. Serial dilutions of treated cells were then plated on YEG media, and colonies counted after 7 days of growth at room temperature. A negative control for both strains included re-suspending cells in 1 M sorbitol instead of the zymolyase solution and repeating the identical protocol otherwise. Percent survival was calculated by dividing the average number of CFU in the zymolyase-treated conditions by the average number of colonies in the sorbitol negative control.

**Dessication sensitivity**. White and opaque cells were grown to the log phase (OD$_{600}$ of 0.5) in YEG media. Two hundred microlitres of culture were harvested, and washed once in 1 ml of dH$_2$O, and pelleted. The pellet was then allowed to dry in a culture hood for 48 h by leaving the tube lids open. As a negative control, cells were re-suspended in 200 µl of dH$_2$O (instead of drying) and left in the same tube rack with the lids closed. After 48 h, dried cells were re-suspended in 200 µl of dH$_2$O. Serial dilutions of both the negative control and the treated cells were then plated on YEG media, and colonies counted after 7 days of growth at room temperature. Percent survival was calculated by dividing the average number of CFU in the dessication-treated tubes by the average number of colonies in the non-dried negative control.

**Growth on Calcfluor white and Congo red**. White and opaque cells were grown to the log phase (OD$_{600}$ of 0.5) in YEG media. Cells were then plated in serial ¼ dilutions onto YEG, YEG + 60 µg/ml calcofluor white, or 0.3 mg/ml Congo red, and incubated at room temperature for 7 days prior to imaging.

**Strain construction**. All DNA manipulations were performed using standard techniques as previously described[48]. Deletion of the EFG1 homologue was accomplished using a two-step method. The EFG1 ORF and 1 kb of flanking sequence was cloned into pUC19 as a *Kpn*1–*Bam*HI fragment using the primers CB299 and CB300. The entire *EFG1* ORF was replaced in this plasmid by the KanMX4 cartridge by amplification of the backbone of the plasmid using the primers CB301 and CB302 and the insert KanMX4 using the primers CB303 and CB304 yielding PCR products with 25 bp overlaps. Following *Dpn*1 digestion, these fragments were mixed 1:1 and added to a Gibson assembly mixture (NEB), followed by incubation at 50 °C for 1 h. A sample of this mixture was transformed into *E. coli* DH5α, and correct assembly was verified by colony PCR using the primers CB305 and CB25, followed by sequencing. For transformation into *T. microellipsoides*, the result plasmid was linearized with *Kpn*I and *Bam*HI, and 1 µg was transformed into white cells using a standard lithium acetate transformation protocol[49], with minor modifications. In place of a heat-shock step, transformation mixtures were incubated overnight at room temperature. Following this, cells were pelleted, re-suspended in YEG and incubated for 8 h at 23 °C prior to plating on selective media. Knockouts were verified by colony PCR using an internal primer set CB280 and CB281 and a primer set CB307 and CB25 confirming correct insertion of the KanMX cartridge. All primers sequences are given in Supplementary Table 2.

**Heat-shock sensitivity**. White and opaque cells were grown to the log phase (OD$_{600}$ of 0.5) in YEG media. Two hundred microlitres of culture were harvested and washed once in 1 ml of dH$_2$O. Cells were then re-suspended in 1 ml of dH$_2$O and divided into two 500 µl tubes. One tube was placed in a 50 °C water bath for 2 min, whereas the other was left on the bench. Serial dilutions of untreated and heat-treated samples were plated, and CFU/ml was then calculated. The percent survival was calculated as the ratio of CFU after heat treatment divided by the CFU in the untreated control.

**RT-qPCR analyses**. RNA was extracted from log or stationary-phase cultures using a hot acid phenol method as previously described[43]. RNA was then treated with DNaseI for 1 h at 37 °C, followed by a single phenol–chloroform extraction to remove DNase activity. One microgram of RNA was reverse transcribed into cDNA using the iScript Reverse transcription supermix kit for RT-qPCR (BioRad), which uses a mix of poly dT and random hexamers for priming. cDNA was quantified using primers to target genes and normalized to levels of histone H3. Primers targeting the following genes were: *CTT1* (CB493 and CB494), *GAP1* (CB495 and CB496), *GLK1* (CB497 and 498), *MEP2* (CB499 and CB500), *SSA3* (CB501 and CB502), *RPL30* (CB503 and CB504), *SOD2* (CB507 and CB508), *XRN1* (CB510 and CB511), and *H3* (CB276 and CB277). The PowerUp SYBR Green Master Mix (Applied Biosystems) qPCR master mix was used for all runs and quantification was determined using an Applied Biosystems Step One Plus Realtime qPCR machine (Applied Biosystems). Data are displayed as delta Ct of the target gene relative to H3.

**Ploidy analysis by flow cytometry**. Cells were grown in 5 ml YEG cultures overnight at room temperature on a roller drum. One hundred microlitre samples of cells were taken and washed twice in 50 mM sodium citrate (pH 7.2). Cells were fixed for 2 h in 95% ethanol, followed by two more washes in 50 mM sodium citrate. Cells were then treated with 1 mg/ml RNaseA in 50 mM sodium citrate for 2 h at 50 °C. SYTOX Green nucleic acid stain (Invitrogen) was then added at a 1/1250 dilution and incubated for 1 h. Samples were analysed using an Attune NxT Flow Cytometer on the FITC channel. Ploidy was estimated as a comparison to white input cells, which are haploid.

**Statistics and reproducibility**. Unpaired *t*-tests were used to compare the means of white and opaque samples in all experimental conditions, and in treated versus untreated samples in test conditions where stressors were added. Raw experimental

measurements and p values are given in Supplementary Data 1. All experiments were reproduced at least three independent times.

**Reporting summary**. Further information on research design is available in the Nature Research Reporting Summary linked to this article.

## Data availability

All raw values for Figs. 1–5, Supplementary Fig. 3, Supplementary Fig. 7, Supplementary Fig. 8, Supplementary Fig. 10, and Supplementary Fig. 12 are given in Supplementary Data 1. An un-cropped gel image of Fig. 1f is shown in Supplementary Fig. 13. Raw sequence data has been deposited on GEO (GSE136991). All other data (if any) are available upon request via email with the corresponding author.

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

## Acknowledgements

We thank Prof. John Smit for his advice and support, Dr. Hao Ding, Dr. Ye Wang and Dr. Christopher Snowdon for their advice and comments on the manuscript, and Rafaela Marodin for technical support with flow cytometry analyses. We also thank Dr. Vivian Miao for assistance with and use of the CHEF gel apparatus for chromosome visualization, and useful comments on the manuscript. C.B. was supported by a Mitacs Elevate Post Doctoral Fellowship (IT09385).

## Author contributions

C.B.: conceptualization, formal analysis, investigation and writing. T.S.: formal analysis, investigation and data curation. M.D. conceptualization, formal analysis, resources, supervision and writing.

## Competing interests

The authors declare no competing interests.
