## [Peer Review File · Communications Biology]

Reviewers' comments:

Reviewer #1 (Remarks to the Author):

This paper describes the first characterization of a phenotypic switch in *T. microellipsoides*: typical small white colonies switch to larger, darker and flatter colonies with an opaque texture. Similar to the white-opaque switch documented exclusively in *C. albicans*, *Candida tropicalis*, and *Candida dubliniensis*. In *T. microellipsoides*, the switching phenotype was unique compared to the *C. albicans* white to opaque transition in terms of transcriptome profiles and cellular DNA content. It was found that the two different morphologies of *T. microellipsoides* have different properties in terms of growth rate, transcriptome, stress tolerance, and genetic stability. The finding that phenotypic switching with highly divergent cell types exists in at least one species in the ZT clade, indicates that morphological transitions occur in more yeasts than had been previously considered.

This paper contains relevant observations, since it concludes that switching is brought about through changes in gene expression patterns. It has long been appreciated that changes in gene expression patterns play a fundamental role in the evolution and establishment of biological diversity.

a) the switching observed in *T. microellipsoides* alters ploidy and the opaque cells are 2-4 fold increased DNA content as compared to that of white cells. This change in ploidy is not observed in the well known morphological changes observed in *Candida albicans*.

b) It was also observed that the switch is reversible (white.opaque and opaque-white) and there is an important increase in genotypic and phenotypic heterogeneity in the population during the switch. I consider this observation relevant since it puts emphasis in the fact that this switch is related to the production of biological novelty. It could be considered that the evolution of transcriptional rewiring could include mechanisms (ploidy) as those observed in this study.

c) The mechanisms behind switching does not seem to be genetic.

As well as in *Candida*, white cells seem to be the basic cell type found in both yeasts. In *T. microellipsoides* white cells are prepared to respond to various stress conditions, while opaque cells are primed for proliferation. Thus switching plays a fundamental role in the physiological adaptation to different environmental conditions.

d) This study opens new insight in regard to the environmental response to stress.

e) The observed transcriptional and phenotypic data clearly indicate that white and opaque cells may be transcriptionally rewired to acquire a more robust or specialized oxidative stress or growth responses, Most important, the analysis of the transcriptome variations in the observed morphological switch, could uncover new transcriptional modulators that could even be acting in peculiar combinations.

Reviewer #2 (Remarks to the Author):

The manuscript by Brimacombe describes genetic mechanisms of phenotypic switching in an

environmental yeast, *Torulaspota microellipsoides*. The manuscript summarizes a very large body of work and documents various genetic, physiological and morphological aspects of switching. The experiments are scientifically sound, and the manuscript is well written. However, the authors should better explain the rationale for this study. The white/opaque switching has been well described in *Candida* species, and similar types of phenotyping switching have been observed in other fungi. Yes, I agree, it has never been documented in this fungus, but why is this important? Better explanations are needed

Please see below responses to reviewer comments in **BOLD**.

Referee expertise:

Referee #1: Yeast metabolism

Referee #2: Fungal diseases

Reviewers' comments:

Reviewer #1 (Remarks to the Author):

This paper describes the first characterization of a phenotypic switch in *T. microellipsoides*: typical small white colonies switch to larger, darker and flatter colonies with an opaque texture. Similar to the white-opaque switch documented exclusively in *C. albicans*, *Candida tropicalis*, and *Candida dubliniensis*. In *T. microellipsoides*, the switching phenotype was unique compared to the *C. albicans* white to opaque transition in terms of transcriptome profiles and cellular DNA content. It was found that the two different morphologies of *T. microellipsoides* have different properties in terms of growth rate, transcriptome, stress tolerance, and genetic stability. The finding that phenotypic switching with highly divergent cell types exists in at least one species in the ZT clade, indicates that morphological transitions occur in more yeasts than had been previously considered.

This paper contains relevant observations, since it concludes that switching is brought about through changes in gene expression patterns. It has long been appreciated that changes in gene expression patterns play a fundamental role in the evolution and establishment of biological diversity.

a) the switching observed in *T. microellipsoides* alters ploidy and the opaque cells are 2-4 fold increased DNA content as compared to that of white cells. This change in ploidy is not observed in the well known morphological changes observed in *Candida albicans*.

b) It was also observed that the switch is reversible (white.opaque and opaque-white) and there is an important increase in genotypic and phenotypic heterogeneity in the population during the switch. I consider this observation relevant since it puts emphasis in the fact that this switch is related to the production of biological novelty. It could be considered that the evolution of

transcriptional rewiring could include mechanisms (ploidy) as those observed in this study.

c) The mechanisms behind switching does not seem to be genetic.

As well as in *Candida*, white cells seem to be the basic cell type found in both yeasts. In *T. microellipsoides* white cells are prepared to respond to various stress conditions, while opaque cells are primed for proliferation. Thus switching plays a fundamental role in the physiological adaptation to different environmental conditions.

d) This study opens new insight in regard to the environmental response to stress.

e) The observed transcriptional and phenotypic data clearly indicate that white and opaque cells may be transcriptionally rewired to acquire a more robust or specialized oxidative stress or growth responses, Most important, the analysis of the transcriptome variations in the observed morphological switch, could uncover new transcriptional modulators that could even be acting in peculiar combinations.

We thank the reviewer for these positive comments.

Reviewer #2 (Remarks to the Author):

The manuscript by Brimacombe describes genetic mechanisms of phenotypic switching in an environmental yeast, *Torulasporea microellipsoides*. The manuscript summarizes a very large body of work and documents various genetic, physiological and morphological aspects of switching. The experiments are scientifically sound, and the manuscript is well written. *However, the authors should better explain the rationale for this study.* The white/opaque switching has been well described in *Candida* species, and similar types of phenotyping switching have been observed in other fungi. Yes, I agree, it has never been documented in this fungus, but why is this important? Better explanations are needed

We apologize for the lack of clarity in this area. Several additional descriptions of the rationale and relevance have been added to the main text. These edits are highlighted within the revised text.

The following specific points are now made:

1. The discovery of this white to opaque-like switch in *T. microellipsoides* was serendipitous; we were studying this fungus for other reasons and observed a phenotype similar to the white to opaque switch in *C. albicans*, and decided to characterize it because of its similarity to the *C. albicans* switch.

2. We now highlight that white to opaque-like switching was thought to be exclusive to related CTG-clade *Candida* species, however we show this general phenomenon is not unique to this clade because something similar exists in *T. microellipsoides*. Morphologic switches in fungi are indeed common (e.g. yeast – hyphae), however white to opaque-like switches from one budding yeast form to a very different budding yeast form have not been documented outside of *Candida* species. We feel this is important because it highlights two points: a) white to opaque-like morphology switching is

unlikely to represent an adaptation to mammalian hosts, as has been proposed in *Candida* literature and b) this variety of phenotypic switching may be a general phenomenon in budding yeast as a mechanism of handling stress/specialized growth conditions, and other uncharacterized yeasts may have similar programs. Our large body of data will certainly facilitate future studies of this phenomenon in this and other yeasts.